# Finite Element Analysis of Electrostatic Coupling in LISA Pathfinder Inertial Sensors

**DOI:** 10.3390/s24196189

**Published:** 2024-09-24

**Authors:** Wenyan Zhang, Jungang Lei, Zuolei Wang, Cunhui Li, Shijia Yang, Jian Min, Xuan Wen

**Affiliations:** Science and Technology on Vacuum and Physics Laboratory, Lanzhou Institute of Physics, Lanzhou 730000, China

**Keywords:** LISA Pathfinder, FE analysis, electrostatic noise, stray electrostatics, residual charge

## Abstract

In the LISA Pathfinder (LPF) mission, electrostatic noise can reach the femto-Newtonian level despite the fact that the LPF’s sensors are equipped with potential shielding. Most of the existing simulation studies focus on the electrostatic edge effect and related fields, while the simulation study of the patch effect is neglected. For that reason, this paper analyzes the basic principle of electrostatic noise and constructs a simulation model for studying the coupling effects of a TM’s residual charge and stray bias voltage. The patch effect and other perturbation factors are simulated by the simulation model with finite element operation, focusing on the suppression effect of the protective ring on the edge effect, the realization of the patch effect in the simulation model, and the possible influence. The results show that electrode area and the spacing between the electrode and the TM together limit the suppression effect of the protective ring on the edge effect. The spatial and temporal variations of the patch effect significantly affect the distributed electric field between the electrodes and the TM, as well as the charge distribution density of the TM. In the worst-case scenario of LPF electrostatic input parameters, the electrostatic noise is about 1.03 × 10^−15^ m/s^2^/√Hz at 1 mHz, which is about 6% different from the expected performance estimate. Finally, considering the limitations of multiple environmental factors on the inertial sensors, the present model will be useful to explore the interactive effects of multi-field coupling and to further investigate the impact of low-energy electron charging on the performance of the inertial sensors.

## 1. Introduction

In the process of space gravitational wave detection, due to extremely weak gravitational wave signals, a commonly used method is to construct a large-scale laser interferometer in space using satellite formations or constellations. Inertial sensors are employed to convert gravitational wave signals into distance change signals between test masses (TMs), and then these distance change signals are read out using a high-precision laser interferometer [1]. The core technology of space gravitational wave detection involves high-precision laser interferometry measurements on a highly accurate space inertial reference platform. Therefore, in order to reach the demand for the detection of gravitational waves on the order of 10^−21^, the LISA mission requires that the residual acceleration of the inertial sensors should be better than 3 × 10^−15^ m/s^2^/Hz^1/2^ at 0.1 mHz.

The European Space Agency (ESA) team has successfully launched LISA Pathfinder (LPF) and has initiated a series of physics experiments to establish the performance budget for the LISA mission. In the physics experiments of LISA Pathfinder [2,3,4,5], the perturbed acceleration noise (PSD^1/2^) experienced by TMs can be obtained by measuring the power spectral density (PSD) of the parasitic force. These experiments provide a detailed summary of the performance metrics for LISA Pathfinder and an estimation of its expected performance. The predicted results are shown in Figure 1 [5].

The LPF mission has relaxed the residual acceleration requirements for TMs to 3 × 10^−14^ m/s^2^/Hz^1/2^ at 1 mHz. As illustrated in Figure 2, reference [6] presents the estimated values of key noise sources from physical experiments at 1 mHz. In LPF, the primarily source of electrostatic interference on TMs is the interaction between the residual charge on the TMs and the stray electrostatic field residual on the GRS electrodes, and the main phenomena involved are the edge effect and the patch effect. The predicted value of the noise due to electrostatic coupling is approximately 1.1 × 10^−15^ m/s^2^/Hz^1/2^ at 1 mHz [6,7,8,9,10,11,12,13].

At present, there is a solid theoretical foundation for electrostatic noise, and numerous finite element simulation studies have been conducted on this phenomenon. Existing simulation studies primarily focus on analyzing the electrostatic edge effect and related aspects, while in contrast, there are relatively fewer studies on the simulation of the patch effect in electrostatic noise. In this paper, we constructed a coupling model of a TM’s residual charge and spurious bias voltage and performed finite element simulations to analyze the edge effect, the patch effect, and other disturbing factors [14].

## 2. Theoretical Analysis

### 2.1. Electrostatic Noise in LPF

In the LPF mission, despite the inherent design of the sensors as equipotential shields, electrostatic noise is still influenced by multiple factors, which can result in electrostatic forces on the order of femtonewtons (*fN*) [15,16,17].

Firstly, TMs in free fall can accumulate charge due to cosmic and solar particle interactions [18]. Secondly, there are stray voltage differences between various points on a gold surface that arise from varying degrees of crystal plane exposure and surface contamination [19]. In LISA Pathfinder inertial sensors, the electrostatic force acting on TMs in a unidirectional manner can be expressed as follows [20,21]:(1)$$F_{Q}=\frac{Q}{C_{t}}\sum \frac{\partial C_{i}}{\partial x}\Delta V_{i}$$

The electrostatic noise experienced by TMs is primarily caused by two factors: The first factor is the coupling of randomly varying charge noise with stray distributed potential differences. The second factor is the coupling of stray potential differences from “patch” potentials or voltage fluctuations in the drive circuit with the average charge [22]. The disturbance noise resulting from both factors can be expressed as follows [21,22,23]:(2)$$\delta F_{Q}=\frac{1}{m_{P}}\frac{\delta Q}{C_{t}}\sum \frac{\partial C_{i}}{\partial x}\Delta V_{i}+\frac{1}{m_{P}}\frac{Q}{C_{t}}\sum \frac{\partial C_{i}}{\partial x}\delta V_{i}$$

In this context, *F_Q_* represents the electrostatic force acting on the TM, *Q* is the residual charge on the TM, *C_t_* is the total capacitance formed between the TM and the surrounding electrodes, ∆*V_i_* is the stray potential difference between the TM and the electrodes, *m_p_* is the mass of the TM, and $\frac{\partial C_{i}}{\partial x}$ denotes the partial derivative of capacitance with respect to the x-gradient.

### 2.2. Stray Distributed Potential

The stray distributed potential has two primary sources: stray electrostatics induced by the driving voltage and stray electrostatics resulting from the patch effect.

#### 2.2.1. Stray Electrostatics Induced by the Driving Voltage

As evidenced by related studies [18,24], the measurement requirements at frequencies of 1 mHz and below necessitate that various circuit components possess extremely high low-frequency stability. Furthermore, the greatest contribution to electrostatic noise arises from the instability of the driving voltage. Low-frequency fluctuations in the amplitude of the driving voltage couple with the accumulated charge on TMs, resulting in the generation of acceleration noise induced by electrostatic drive along the *x*-axis. In accordance with the specifications of LPF’s inertial sensors, the requirement for driving voltage noise is approximately 10 μV/√Hz at 1 mHz.

#### 2.2.2. Stray Electrostatics Induced by the Patch Effect

Other potential sources of stray electrostatic noise in LPF are related to the patch effect and surface contamination. In an ideal geometric model, each electrode should exhibit a uniform spatial potential. However, in practice, variations in exposed crystal planes, work function, and surface contamination of the metal can lead to the emergence of stray potential differences between different points on the conductive surface [25].

The electrostatic perturbations induced by the patch effect primarily manifest in two aspects: the spatial difference between the TMs’ and electrodes’ surface potential, and the temporal variations of the electrodes’ surface potential. The spatial variations can lead to the force gradient, which causes acceleration noise. On the other hand, the temporal variations of the electrodes’ surface potential interact with the DC voltage present in the environment or the net free charge on the test mass, which may also generate acceleration noise [26]. According to literature [27,28], the spatial variation of the patch effect should not exceed 100 mV on the centimeter scale, and the estimated value of the temporal fluctuation of the electrodes’ surface potential is approximately 100 μV/√Hz at 1 mHz due to the patch effect.

In order to study the potential distribution on the surface of a centimeter-scale macroscopic conductor in a high-precision space gravity experiment, Stanford University measured the potential distribution on the surface of a centimeter-scale test mass using a classical Kelvin probe with a millimeter-scale tip. The measurement results are shown in Figure 3, which illustrates the potential distribution on the surface of the corresponding conductor [20].

### 2.3. Residual Charge on the TM

In the LPF mission, high-energy particles, including those originating from galactic and solar sources, can easily penetrate the spacecraft’s structure, resulting in the charging of the freely floating TM. As demonstrated in studies such as those presented in [28,29], accurate simulations of a TM’s charging behavior have been conducted using the Geant4 toolkit in conjunction with comprehensive geometric and physical models. The results indicate that under conditions of solar maximum activity, the charging rate of a TM approaches 25 +e/s, while under solar minimum conditions, the charging rate increases to 50 +e/s.

## 3. Finite Element Simulation Analysis

### 3.1. Geometric Model

In gravitational wave detection missions, the geometric design of TMs and electrodes has a direct impact on the performance and accuracy of gravitational wave detectors [9,30]. In order to constrain the design of the main interfaces for the sensitive probe, we analyzed the factors influencing residual acceleration based on the working principles of inertial sensors. Furthermore, in consideration of the requisite specifications for inertial sensors to achieve triaxial six-degrees-of-freedom displacement sensing and electrostatic drive control, the TM was constructed as a 46 mm cube comprising 73% gold (Au) and 27% platinum (Pt) materials, with a gold-plated surface and a weight of 2 kg [31]. The EH-compatible geometry of the TM was also cubic in structure and was composed of molybdenum with a gold-plated finish [32]. A light aperture was designed in the direction of laser interference to measure the distance between TMs. The *x*-axis served as the direction of laser interference, and to reduce the effects of stiffness coupling, no injection electrodes were placed in this direction. In contrast, injection electrodes were located in the non-laser interference directions [33]. The EH included 12 sensing (or actuation) electrodes and 6 injection electrodes arranged in a centrally symmetric configuration. The sensing/driving electrodes were distributed on both sides of the TM to increase the moment arm length of the electrostatic force. Additionally, a grounding protection ring was designed with the intention of limiting the edge effects of the electric field. The specific configuration is shown in Figure 4.

The characterization of the DC voltage offsets between the TM and the electrodes was complicated by the presence of differences in work function, non-uniformity, and contamination. These can increase risks in space experiments and generate parasitic electrostatic stiffness proportional to d^−3^ [29,34]. In order to mitigate the influence of electrostatic stiffness, the sensing gap for the sensitive *x*-axis was designed to be 4 mm. Furthermore, in consideration of the sensitivity requirements for translational and rotational positions, the sensing gaps for the *y*-axis and *z*-axis were designed to be 2.9 mm and 3.5 mm, respectively, when considering the area ratio of the sensing/driving electrodes to the injection electrodes [5]. The injection electrodes were designed on the y and z planes and were slightly recessed with gaps of 4 mm between them in order to reduce the electrostatic stiffness associated with measurement offsets.

Based on the aforementioned design requirements, a finite element geometric model was constructed, as illustrated in Figure 5, with the relevant material parameters enumerated in Table 1 [35].

### 3.2. Simulation Model

By analyzing various types of perturbation factors in the electrostatic noise of inertial sensors, we found that the dielectric medium between the TM and the EH didn’t involve current flow, and the main sources of the electric field were the stray potential and charge distribution. Under the electrostatic physical field interface, we simulated the coupling effect between the stray potential and the residual charge of the TM by setting boundary conditions. The model simulation was then conducted, and the finite element simulation flowchart was constructed, as shown in Figure 6.

Based on the studies presented in Section 2.2.1 and Section 2.2.2, and considering the worst-case electrostatic perturbation, we constructed a simplified approximate model as shown in Figure 7. The model assumed that the patch effect acts on a single sensing electrode along the *x*-axis, which results in both spatial and temporal variations in the electrode surface potential. Based on the distribution scale for domain cuts on the electrode surface, shown in Figure 3, the voltage was applied to each cut domain with an area of ∆a (≈1 mm^2^) as illustrated in the figure. The voltage of each domain was spatially variable and limited to a range of 100 mV, with fluctuations over time of about 100 μV/√Hz at 1 mHz. Furthermore, in addition to the DC offset resulting from the patch effect, low-frequency sinusoidal driving voltage fluctuations also existed, which were set to 10 μV/√Hz at 1 mHz [14,26,27,28]. The maximum charging rate of the TM was considered to be 50 +e/s in accordance with the results obtained from the Geant4 model. This ensured that the residual charge on the TM did not exceed 1 × 10^−13^ C during the simulation period [29].

## 4. Results and Discussion

In order to achieve an optimal balance between computational accuracy and computation time, the geometric model was divided into finite element meshes as shown in Figure 8. The mesh in the electrostatic model was of greater density at critical locations, such as corners and junctions, while ensuring a constant and uniform mesh distribution within the plane.

### 4.1. Edge Effect and Grounding Protection Ring

As shown in Figure 9, the design of capacitors typically employs equipotential technology to ensure a uniform distribution of the electric field between the measuring electrode and the target conductor. This is achieved by adding an equipotential protection ring at the periphery of the pole plate. This design can significantly improve the sensitivity and anti-interference capability of a capacitor while also effectively redirecting the edge effects of the measuring electrode outside the protective ring, thereby reducing the impact of these edge effects on the test results [14,34,35].

In the design of sensitive probes for inertial sensors, a protective ring is formed by grounding the area between the sensing electrode and the injection electrode to effectively suppress the effects of direct coupling between the electrode surfaces. This design aims to ensure that the electric field lines induced by the electrodes terminate at their respective TM surfaces rather than at other electrodes or neighboring TM surfaces [36], as shown in Figure 10. This arrangement not only limits the leakage of the electric field, but also reduces the cross-coupling effect of the forces, thereby improving measurement sensitivity.

The electrodes are electrically isolated from the structure of the housing with ceramic bushings, and the electrode cages form a complete Faraday cage structure with the housing to ensure that the potential signals between the TM and the electrodes are not affected by external electromagnetic interference [37]. Furthermore, to avoid unwanted interference from the insulating spacers on the TM, these spacers should be placed within the coverage of the electrodes, ensuring that they are not exposed to the TM’s field of view, thus maintaining the reliability of the signal during the measurement process [38].

In the design of inertial sensors, the edge effect is an important factor to consider. Ideally, the edge effect disappears when the electrode area is infinitely large and the spacing between the TM and the electrode is infinitely small. According to the design parameters of LPF inertial sensors, the spacings between the TM and the sensing electrodes are 4 mm, 2.9 mm, and 3.5 mm in the x, y, and z directions, respectively, while the areas of the individual sensing electrodes are 504 mm^2^, 266 mm^2^, and 252 mm^2^, respectively. The distribution shown in Figure 11 was obtained by simulating the electric field between the TM and the sensing electrodes in the x, y, and z directions.

The results showed that the protection ring exhibited varying effects in suppressing the edge effect between the sensing electrode and the TM, as follows: The strongest suppression effect occurred in the *x*-axis direction, the second strongest in the *z*-axis direction, and the weakest in the *y*-axis direction. Therefore, in the design process of inertial sensors, finite element simulation can be used to make a comprehensive trade-off between electrode area, TM and electrode spacing, and other factors to achieve effective suppression of the edge field, thereby improving the sensitivity of the measurements.

### 4.2. Patch Effect

Referring to the simplified approximation model described in Section 3.2, we performed domain cuts on the surface of a single *x*-axis sensing electrode in the finite element model at the millimeter scale. Specifically, we divided the electrode surface into multiple small-area domains, each with an area of ∆a (≈1 mm^2^). On these cut domains, a spurious bias, as shown in Figure 12, was applied with reference to the spatial difference constraints of the patch effect. At the same time, this voltage varied over time by about 100 μV/√Hz at 1 mHz.

Referring to the driving noise data of the electrostatic suspension accelerometer at the Institute 510, the input stray voltage noise spectral density curve is shown in Figure 10, which shows a fluctuation magnitude of about 100 μV/√Hz at 1 mHz (Figure 13). In addition to the fluctuations of the stray DC offset due to the patch effect, low-frequency sinusoidal driving voltage fluctuations also produced a stray voltage noise spectral density curve, with a fluctuation magnitude of approximately 10 μV/√Hz at 1 mHz (Figure 14).

Figure 15a shows a schematic diagram of the electric field between the TM and the sensing electrodes in the absence of the patch effect. In this case, the maximum value of the electric field was approximately 4.49 V/m. In contrast, Figure 15b presents a schematic diagram of the electric field between the TM and the sensing electrode in the presence of the patch effect at a single electrode in the *x*-axis direction. In this case, the maximum value of the electric field increased significantly to 146 V/m.

Figure 16a illustrates the charge distribution density plot of the TM in the absence of the patch effect, while Figure 16b illustrates the charge distribution density plot of the TM in the presence of the patch effect on a single electrode. The results indicate that the patch effect significantly impacted the charge distribution density of the TM, even though the total amount of charge of the TM remained constant over the same time period. Additionally, Figure 17 shows that the distribution of charge on the surface of the ceramic spacer was also influenced by the patch effect.

Overall, the patch effect significantly increased the electric field between the TM and the sensing electrodes. This increase in the electric field strength not only occurred near the surfaces where the voltage was spatially, but also extended to the electric field in the edge regions of the electrodes. In addition, the distribution of charge on the surface of the ceramic spacer was similarly affected by the patch effect during the operation of the inertial sensors. Therefore, the impact of the patch effect on the electric field and charge distribution must be fully considered when designing inertial sensors to ensure the system’s performance and stability.

### 4.3. Analysis of Capacitance, Electrostatic Force, and Torque

This section analyzes the theoretical calculations and simulation results regarding the capacitance between the TM and the electrodes, as well as the electromagnetic force and the torque acting on the TM. Table 2 shows the theoretical capacitance values derived from the ideal parallel plate capacitor model alongside the capacitance values obtained from the finite element model simulations for both the sensing/driving and injection electrodes. As discussed in Section 3.1, for the *x*-axis sensing/driving electrodes with a relatively large gap between the TM and the electrodes, the simulated capacitance was lower than the calculated value due to the significant suppression of edge field effects by the protection ring. In contrast, for the *y*-axis and *z*-axis sensing/driving electrodes, the infinite parallel plate approximation overlooked the edge field effect, resulting in an inflated effective electric field area that was not accounted for in the theoretical calculations. This led to simulated capacitance values that exceeded the calculated values.

Table 3 shows the values of electromagnetic force and torque obtained from the theoretical calculations and from the finite element simulations. Based on a qualitative analysis of the electrostatic force Equation (1), the simulated values of the electromagnetic force and torque are lower than the theoretically calculated values, which is a reasonable result.

### 4.4. Electrostatic Noise Due to Coupling of Stray Voltage and Residual Charge

Based on the constructed simulation model, when the stray DC offset caused by the patch effect was 100 mV, the stray voltage fluctuation was about 100 μV/√Hz at 1 mHz, and the low-frequency sinusoidal driving voltage fluctuation was around 10 μV/√Hz at 1 mHz. The charging rate for the TM was 50 +e/s while the residual charge on the TM did not exceed 1 × 10^−13^ C during the simulation period. Under these conditions, the electrostatic noise source resulted in a residual acceleration noise of approximately 1.03 × 10^−15^ m/s^2^/√Hz at 1 mHz. The spectral density curve of the electrostatic noise is shown in Figure 18, and this result deviates by about 6% from the expected performance estimate for electrostatic noise at 1 mHz in Figure 2.

## 5. Conclusions

In this study, we focused on the design index of inertial sensors in the LPF space gravitational wave detection mission. A finite element model was used to simulate and analyze the coupling effect between stray voltage fluctuations and the TM residual charge, primarily involving two factors: the edge effect and the patch effect. The following conclusions were drawn:(1)We found through simulation that the protective ring exhibited different effects on different axes in suppressing the edge effect between the electrode and the TM: the *x*-axis direction showed the strongest suppression effect, the *z*-axis direction demonstrated the second strongest suppression effect, and the *y*-axis direction had the weakest suppression effect. This variation also led to differences between the simulated and calculated values of capacitance, electromagnetic force, and torque in the sensitive probe of the inertial sensors.(2)In this study, we investigated the specific implementation of the patch effect in the simulation model, and the simulation results show that the spatial differences and temporal variations of the patch effect correlate with the electric field distribution between the TM and the electrodes, the charge distribution of the TM, and the charge of the ceramic spacer. Furthermore, this correlation impacted the overall performance of the inertial sensors.(3)In this model, the charging rate of the TM was simulated with a constant value under worst-case conditions. However, according to the analysis results from the Geant4 model, the charging rate of the TM can vary with the influence of space environmental factors such as solar activity. Therefore, future research will incorporate charge transport models to investigate the charging and discharging processes of the TM and provide a deeper understanding of how the charging behavior of low-energy electrons affects the performance of the inertial sensor.(4)Although our research primarily focused on the simulation and analysis of electrostatic noise in inertial sensors, we also recognize that other factors such as temperature, pressure, and magnetic field can significantly affect their performance. Therefore, to address this important issue, we plan to adopt a multi-field coupling approach in our future research to explore the interactions among these environmental factors in greater depth. By extending the simulation model to include more environmental variables, we hope to evaluate the performance of inertial sensors under complex environmental conditions more comprehensively. Ultimately, our goal is to develop an integrated model that more closely resembles the noise characteristics of real inertial sensors, thereby providing a theoretical foundation and data support for optimizing sensor performance in practical applications.

## Figures and Tables

**Figure 1 sensors-24-06189-f001:**
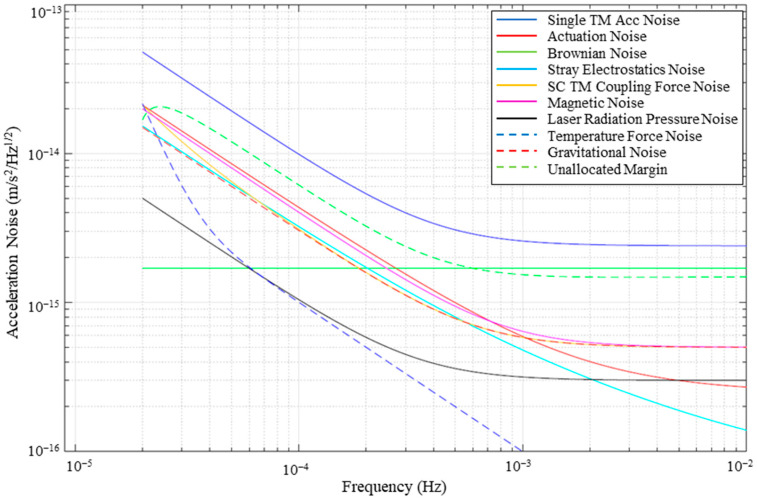
Schematic of LPF’s metric prediction on decomposed noise sources.

**Figure 2 sensors-24-06189-f002:**
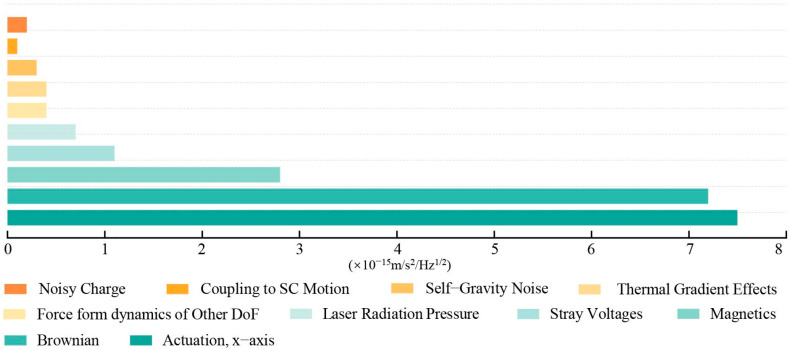
PSD^1/2^ of LPF acceleration noise sources at 1 mHz.

**Figure 3 sensors-24-06189-f003:**
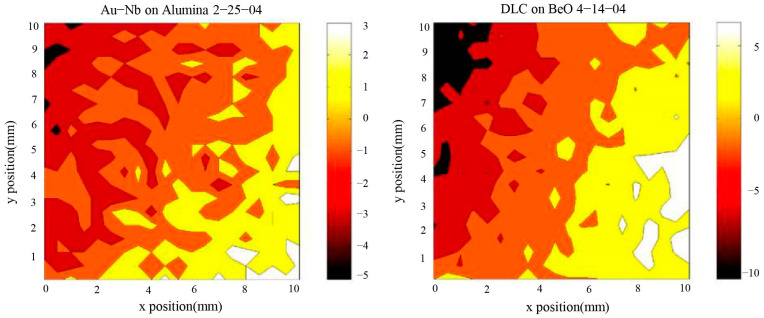
The surface potential distribution of a conductor measured by the Kelvin probe system.

**Figure 4 sensors-24-06189-f004:**
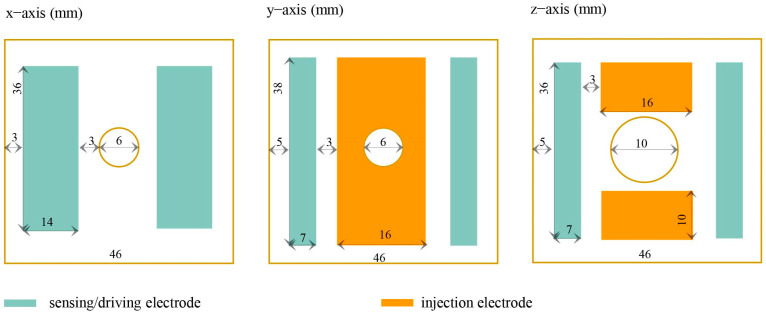
Electrode configuration for GRS.

**Figure 5 sensors-24-06189-f005:**
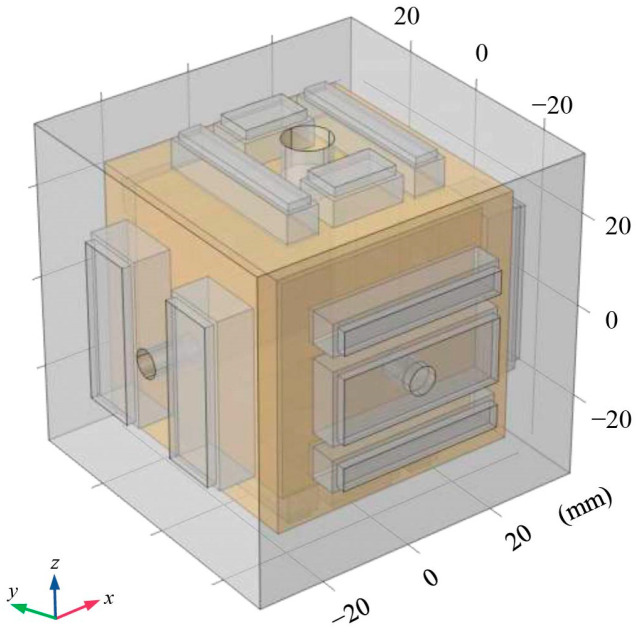
Finite element geometry model.

**Figure 6 sensors-24-06189-f006:**
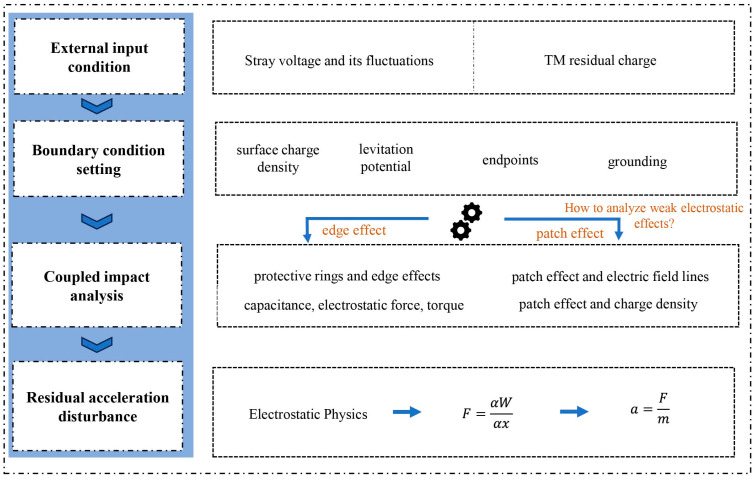
Flowchart for electrostatic noise simulation based on the finite element method.

**Figure 7 sensors-24-06189-f007:**
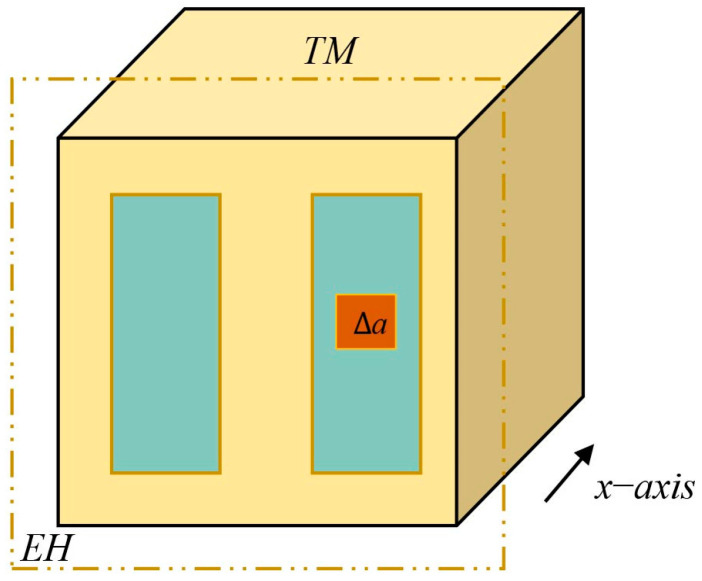
Simplified role model.

**Figure 8 sensors-24-06189-f008:**
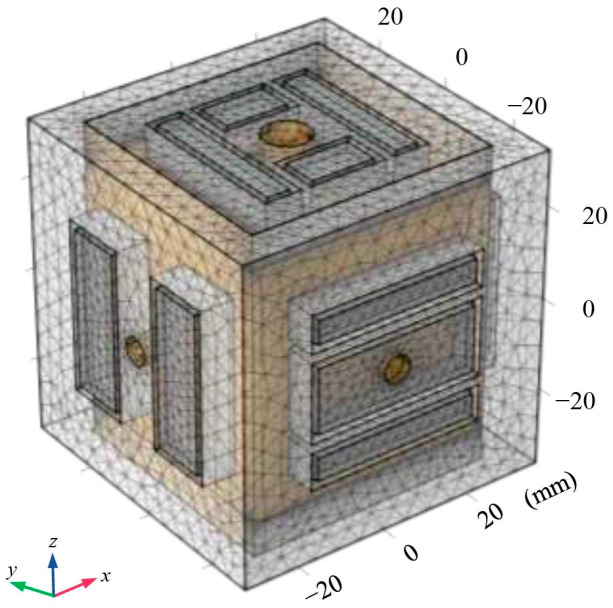
Meshing of finite element geometric model.

**Figure 9 sensors-24-06189-f009:**
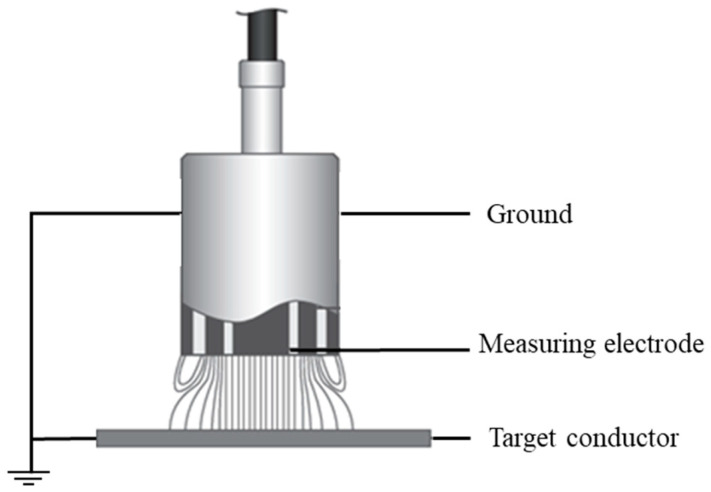
Schematic diagram of the basic principle of grounding protection ring.

**Figure 10 sensors-24-06189-f010:**
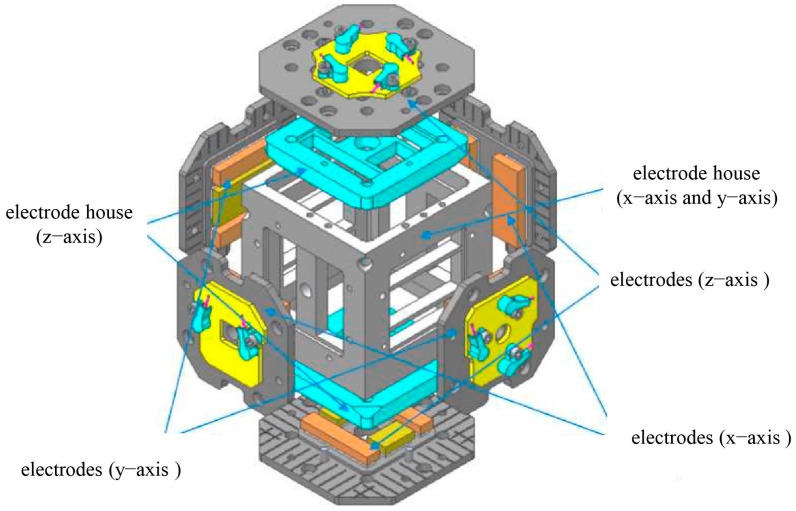
Schematic design of electrode house and electrode structure in inertial sensor.

**Figure 11 sensors-24-06189-f011:**
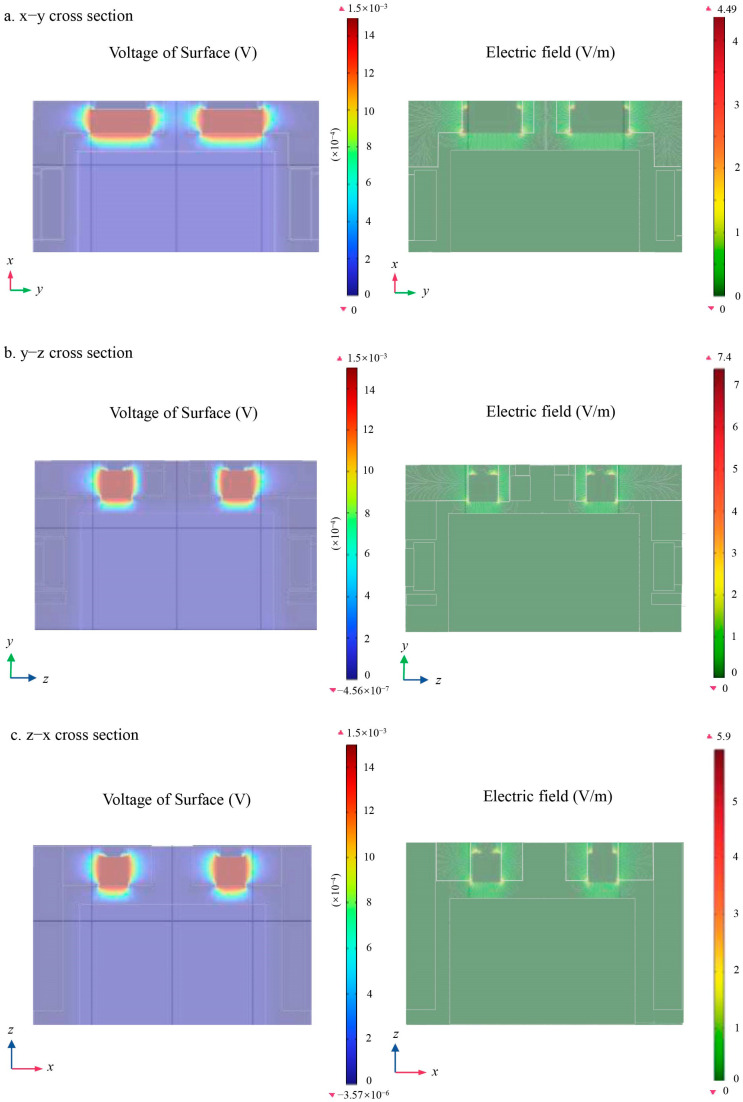
Cross-sections of surface voltage and electric field (**a**) x-y cross-section; (**b**) y-z cross-sections; (**c**) z-x cross-sections.

**Figure 12 sensors-24-06189-f012:**
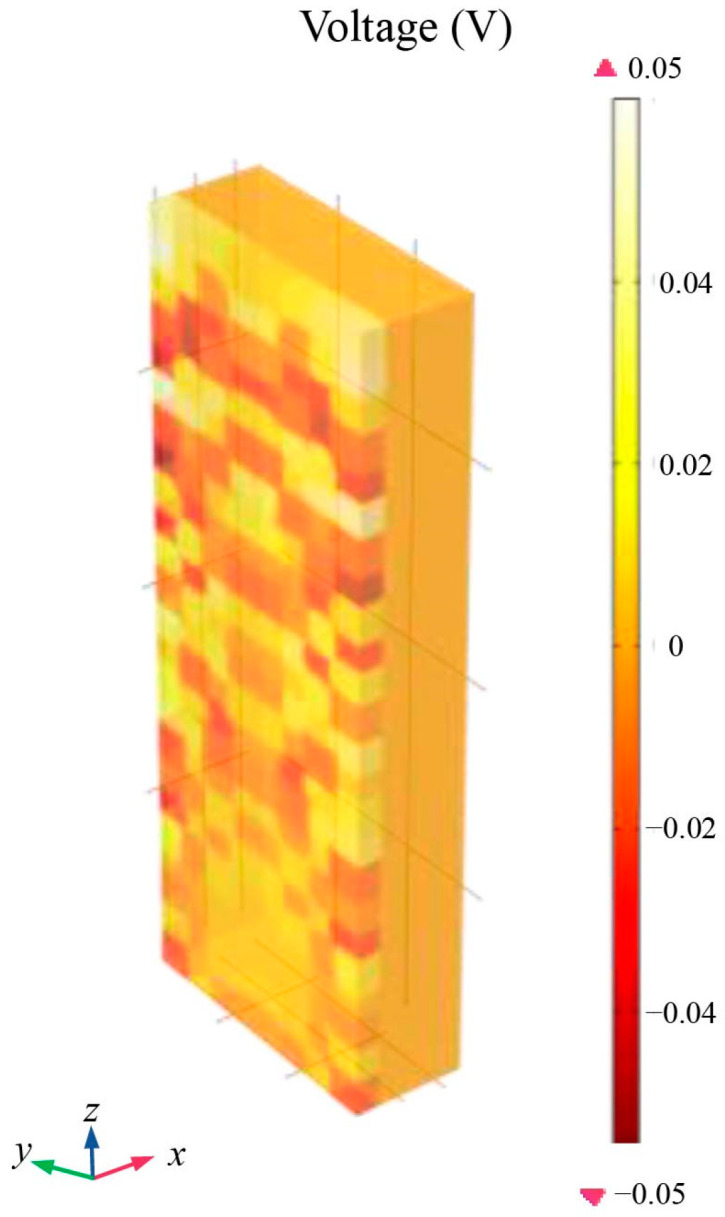
Schematic of the spatial variation of the patch effect.

**Figure 13 sensors-24-06189-f013:**
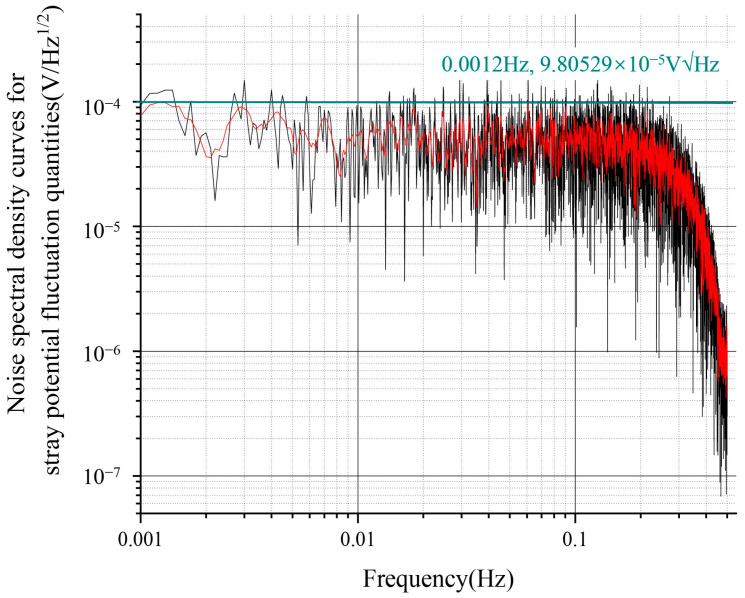
Spectrogram of voltage fluctuations corresponding to the patch effect.

**Figure 14 sensors-24-06189-f014:**
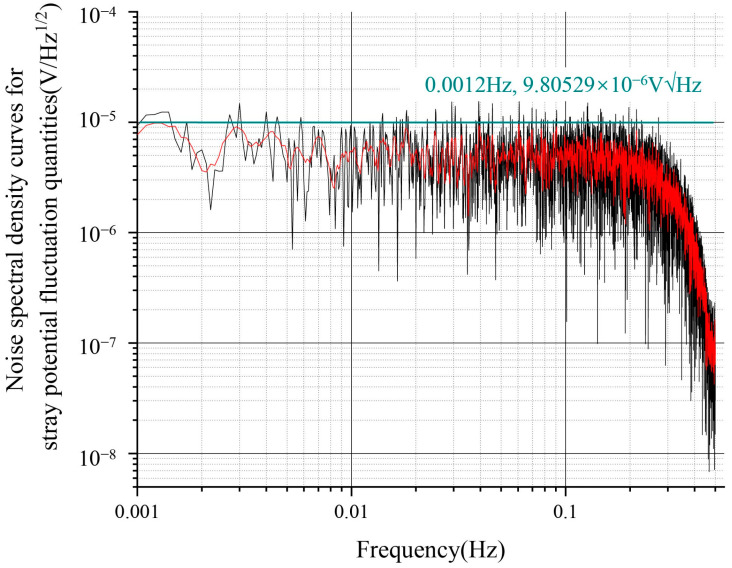
Spectrogram of sinusoidal driving voltage fluctuations.

**Figure 15 sensors-24-06189-f015:**
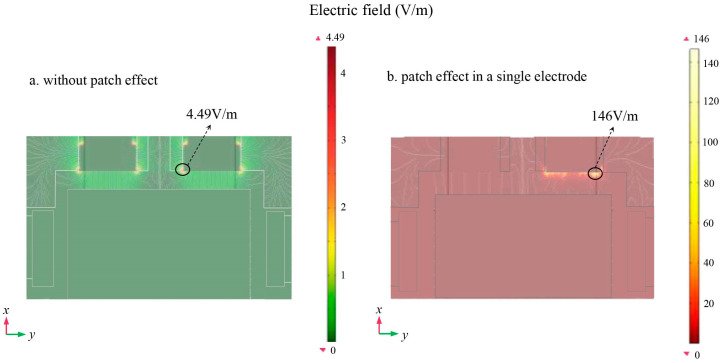
Schematic diagram of the potential mode of the cross section between TM and sensing/driving electrode (**a**) without patch effect; (**b**) patch effect in the *x*-axis.

**Figure 16 sensors-24-06189-f016:**
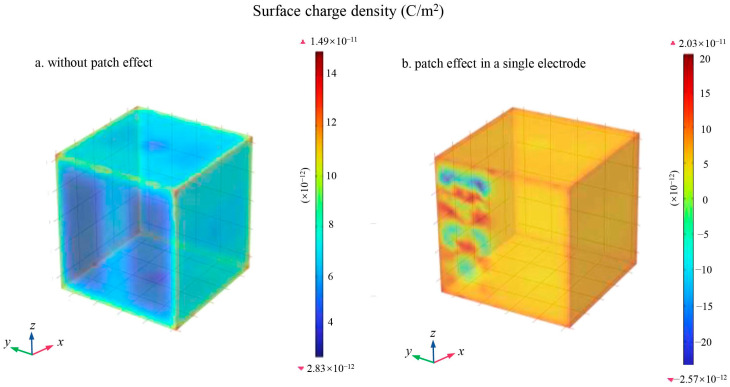
Charge distribution density plot of TM. (**a**) without patch effect; (**b**) patch effect in the *x*-axis.

**Figure 17 sensors-24-06189-f017:**
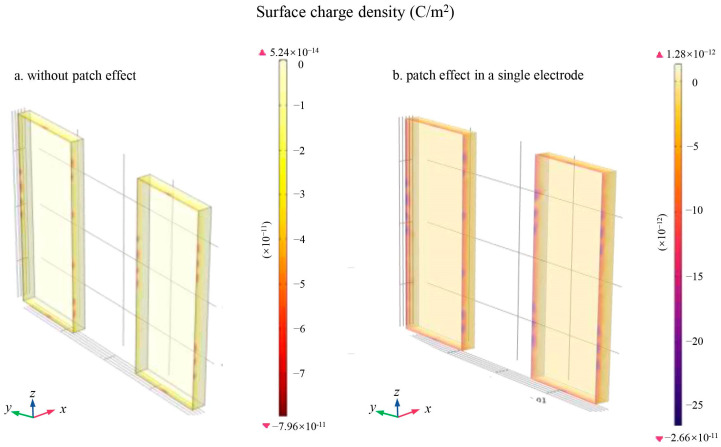
Charge distribution density plot of ceramic spacer layer. (**a**) without patch effect; (**b**) patch effect in the *x*-axis.

**Figure 18 sensors-24-06189-f018:**
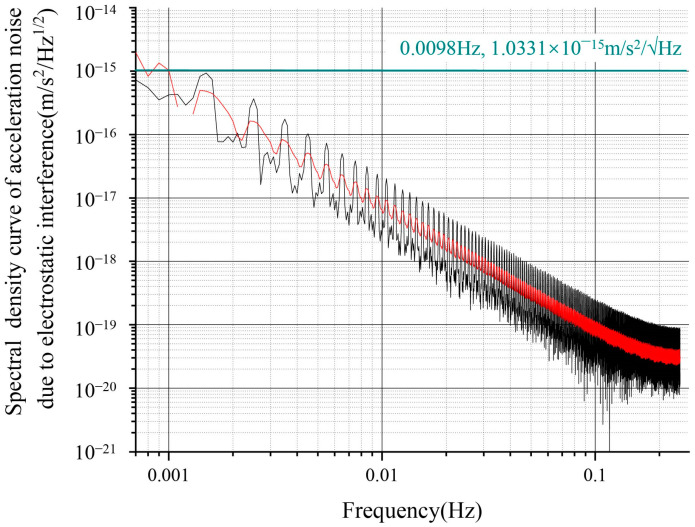
Spectral density curve of acceleration noise due to electrostatic interference.

**Table 1 sensors-24-06189-t001:** The main material parameters of GRS.

Material	Unit	73%Au27%Pt	Gold	Molybdenum	Aluminum Nitride
Density	kg/m^3^	20,547	19,320	10,200	3250
Relative permittivity	1	3	2.4	4.8	9
Poisson’s ratio	1	0.36	0.4	0.31	0.25
Thermal conductivity	W/(m·K)	317	317.9	176	300
Young’s modulus	Pa	150 × 10^9^	79 × 10^9^	329 × 10^9^	300 × 10^9^
Constant pressure heat capacity	J/(kg·K)	127	128	138	750

**Table 2 sensors-24-06189-t002:** Theoretical calculations and finite element simulation results of electrodes.

Capacitance	C_xsens-TM_	C_ysens-TM_	C_zsens-TM_	C_yinj-TM_	C_zinj-TM_	C_EH-TM_	C_total_
Calculated value	1.12 × 10^−12^	8.12 × 10^−13^	6.37 × 10^−13^	1.58 × 10^−12^	3.54 × 10^−13^	1.49 × 10^−11^	2.98 × 10^−11^
Simulated value	1.22 × 10^−12^	9.46 × 10^−13^	8.12 × 10^−13^	1.54 × 10^−12^	3.44 × 10^−13^	2.11 × 10^−11^	3.75 × 10^−11^
Error	8.60%	14.19%	21.53%	2.49%	2.85%	29.28%	25.93%

**Table 3 sensors-24-06189-t003:** Theoretical calculation and finite element simulation results of electromagnetic force and torque.

Type	Axis	Calculated Value	Simulated Value	Error
Force (N)	*X*-axis	5.60 × 10^−12^	4.20 × 10^−12^	25.04%
	*Y*-axis	9.35 × 10^−12^	7.16 × 10^−12^	23.47%
	*Z*-axis	4.51 × 10^−12^	4.19 × 10^−12^	7.11%
Torque (N·m)	Rotation around *z*-axis	5.14 × 10^−14^	4.46 × 10^−14^	13.16%
	Rotation around *x*-axis	7.11 × 10^−14^	5.47 × 10^−14^	23.14%
	Rotation around *y*-axis	5.88 × 10^−14^	5.43 × 10^−14^	7.63%

## Data Availability

The data presented in this study are available on request from the corresponding author.

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
