# Peer review of "Finite Element Analysis of Electrostatic Coupling in LISA Pathfinder Inertial Sensors"

_sensors, 2024, doi:10.3390/s24196189_

Round 1

Reviewer 1 Report

Comments and Suggestions for Authors

This report investigates the electrostatic noise in LPF inertial sensors using finite element simulation modeling. As a design tool for LISA/LPF, the work provides an in-depth analysis and understanding of electrostatic noise in inertial sensors, taking into account the current design advancements. With a clear topic, a well-defined research direction, and thorough analysis of the results, this paper can provide a theoretical and practical reference for the study of electrostatic noise in LPF inertial sensors. However, there are several minor issues that should be addressed:

--- There is redundancy in the abstract section, which could be further optimized.

--- Although the article provides a comprehensive discussion of the finite element simulation and analysis method, the system flow diagram of the finite element model needs to be added to enhance the completeness of the simulation model.

--- References [21,23] need to be updated to include the latest research findings.

--- The English grammar of the entire article should be reviewed to improve the language quality.

--- While the article focuses primarily on electrostatic noise, inertial sensors are also affected by various factors such as temperature and pressure. Therefore, in the future research direction, models for multiple influencing factors can be proposed for simulation and analysis to evaluate the performance of inertial sensors more comprehensively.

Reviewer 2 Report

Comments and Suggestions for Authors

This manuscript reports an FEA for the LISA Pathfinder inertial sensor. I think that this manuscript must be improved with a major revision. The reasons are listed as follow:

1)  The finite element simulation for LPF has been reported detailedly in Ref [2]. For example, the results in table 2 and table 3 are similar with table 2 in Ref [2]. As for the guard ring mentioned in this manuscript, it is a simple and common problem for the capacitor design. Is there any special design for the guard ring?

2) The only difference between this manuscript and Ref [2] is the introduction of patch effect. However, the method used here is adding an area of Delta a over one side. Can this procedure produce the patch potential in the surface of electrode?

Round 2

Reviewer 2 Report

Comments and Suggestions for Authors

I will be not object to publication.